# Predictors of Adherence to Lifestyle Recommendations in Stroke Secondary Prevention

**DOI:** 10.3390/ijerph18094666

**Published:** 2021-04-27

**Authors:** Olive Lennon, Patricia Hall, Catherine Blake

**Affiliations:** Health Sciences Centre, School of Public Health, Physiotherapy and Sports Science, University College Dublin, Belfield, D04 V1W8 Dublin 4, Ireland; patricia.hall@ucdconnect.ie (P.H.); c.blake@ucd.ie (C.B.)

**Keywords:** stroke, secondary prevention, cardiovascular risk, health behaviours, lifestyle, risk reduction

## Abstract

The risk of recurrent vascular events is high following ischaemic stroke or transient ischaemic attack (TIA). Unmanaged modifiable risk factors present opportunities for enhanced secondary prevention. This cross-sectional study (*n* = 142 individuals post-ischaemic stroke/TIA; mean age 63 years, 70% male) describes adherence rates with risk-reducing behaviours and logistical regression models of behaviour adherence. Predictor variables used in the models com-prised age, sex, stroke/TIA status, aetiology (TOAST), modified Rankin Scale, cardiovascular fit-ness (VO_2peak_) measured as peak oxygen uptake during incremental exercise (L/min) and Hospital Anxiety and Depression Score (HADS). Of the study participants, 84% abstained from smoking; 54% consumed ≥ 5 portions of fruit and vegetables/day; 31% engaged in 30 min moderate-to-vigorous physical activity (MVPA) at least 3 times/week and 18% were adherent to all three behaviours. VO_2peak_ was the only variable predictive of adherence to all three health behaviours (aOR 12.1; *p* = 0.01) and to MVPA participation (aOR 7.5; *p* = 0.01). Increased age (aOR 1.1; *p* = 0.03) and lower HADS scores (aOR 0.9; *p* = 0.02) were predictive of smoking abstinence. Men were less likely to consume fruit and vegetables (aOR 0.36; *p* = 0.04). Targeted secondary prevention interventions after stroke should address cardiovascular fitness training for MVPA and combined health behaviours; management of psychological distress in persistent smokers and consider environmental and social factors in dietary interventions, notably in men.

## 1. Introduction

Ischaemic stroke is a leading cause of death and adult-acquired disability [1]. Over 67.5 million people globally are living following an ischaemic stroke event [2]. Secondary prevention is critical as recurrent stroke risk is 6- to 15-fold higher than the risk of stroke in the general population [3,4]. Recurrent stroke is associated with higher mortality rates and increased disability levels [3,5,6,7]. First-ever ischaemic stroke is further associated with an increased risk of incident heart disease [8] and future non-stroke vascular death and myocardial infarct [9,10]. 

Ten modifiable risk factors (all significant for ischaemic stroke) account for 90% of the Population-Attributable Risk of stroke [11]. Epidemiological studies further support the protective effects from stroke for healthy patterns with high fruit and vegetable consumption [12,13,14], smoking abstinence [15], physical activity participation [16,17,18]. Compliance with multiple low-risk lifestyle behaviours relating to smoking, BMI, physical activity, alcohol consumption, and diet reduces the risk of ischaemic stroke by up to 80% [19]. Following a stroke, moderate-level evidence [20] supports the role of a Mediterranean-style diet [21], a dose–response, 2-fold risk of stroke recurrence is identified in smokers [22] and physical activity participation is associated with a 40% lower risk of a recurrent stroke, myocardial infarction or vascular death at 3 years in the SAMMPRIS trial [23]. 

Pathophysiological factors, stroke subtype (aetiology) and lifestyle factors have been used to identify people at risk for recurrent events [24,25,26]. However, longitudinal studies addressing recurrence lack specific focus on lifestyle factors [27,28,29] and some stroke types for example cardioembolic stroke may not be as strongly linked to established behavioural risk factors. Significantly lower engagement in healthy lifestyle behaviours is documented in stroke survivors than in matched controls [30]. Even after a life-threatening vascular event such as stroke, and lifestyle advice as routine practice, one-third of smokers continue to smoke [31,32]; 58% fail to meet guidelines for physical activity (PA) [33] and more than half (55%) have low fruit and vegetable consumption [34]. Physical, environmental and psychological aspects have been cited by people as barriers to engagement in healthy lifestyle after stroke [35]. 

Currently, no data identify characteristics of individuals after stroke who are more likely to adhere to healthy lifestyle guidelines and patterns of combined health behaviours. We hypothesised that those with non-cardiac stroke sources, those with greater stroke-related disability and greater psychological distress will have lower adherence levels. Therefore, the aims of this study are to:(a)Present an a priori conceptual model of potential factors influencing health behaviours after stroke or TIA (Figure 1). These include sociodemographic factors (age and gender), stroke-related factors (TOAST classification, stroke or TIA classification) and stroke sequalae (acquired disability, reduced cardiovascular fitness and psychological distress).(b)Establish unadjusted prevalence rates of adherence with healthy lifestyle behaviours of absolute smoking abstinence, a minimum of 30 min of moderate–vigorous physical activity (MVPA) at least 3 days per week, consumption of greater than or equal to 5 portions of fruit and vegetables per day and adherence with all three behaviours.(c)Identify determinants of each healthy lifestyle behaviour and adherence with all three from those proposed in the conceptual model (Figure 1).

## 2. Materials and Methods

### 2.1. Research Design

A cross-sectional study of baseline data from the Cardiac Rehabilitation adapted for TIA and Stroke (CRAFTS) trial was employed. Full details of the protocol for the CRAFTS trial are published elsewhere [36]. This study received full ethical approval, as detailed in the Institutional Review Board Statement. 

### 2.2. Participants

Participants with ischaemic stroke were recruited from community stroke services and TIA participants were recruited from a tertiary TIA clinic. Inclusion criteria included medically stable individuals post ischaemic TIA or stroke with consent of a vascular neurologist or General Practitioner; over 18 years of age and of either sex. Exclusion criteria included oxygen dependence, unstable angina and other uncontrolled cardiac conditions, unstable diabetes, major medical conditions, symptomatic claudication, febrile illness, pregnancy or cognitive impairment limiting informed consent. Volunteers were screened by phone to ensure they fulfilled the inclusion criteria and no exclusion criteria applied. The Physical Activity Readiness Questionnaire (PAR-Q) was used to screen for exercise testing suitability [37]. 

### 2.3. Health Behaviours

Following informed written consent, subjects completed self-report measures of health behaviours including the short form International Physical Activity Questionnaire (IPAQ) [38], the Food Frequency Questionnaire [39] and a yes/no response item to smoking status followed by the Smoking Ladder Questionnaire for those in the yes category [40]. 

### 2.4. Participant Profile

Baseline descriptives were recorded from participants’ medical records that included stroke or TIA status and aetiological classification using the Trial of Org 10172 in Acute Stroke Treatment (TOAST) system [41]. Participants were medically examined prior to testing. Resting heart rate and blood pressure were measured following a seated rest period of 10 min by the medical officer. Body mass and stature were measured using a calibrated weighing scale and stadiometer (SECA Medical 703 Digital Column Scale, Hamburg, Germany) and body mass index (BMI) calculated according to the formula (body mass/stature^2^).

### 2.5. Stroke-Related Issues

Participants completed the Hospital Anxiety and Depression Questionnaire [42] where Depression and Anxiety scores were summed to provide a bifactor model of psychological distress [43]. Modified Rankin Scale scores (mRS) [44] were assigned by the medical officer, identifying stroke severity, where zero indicates no disability and 5 indicates total dependence for all self-care and mobility. Functional ambulatory category scores (FAC) were assigned indicating mobility status where zero indicates someone who is non-ambulatory and 5 indicates independent mobility on all ground surfaces and steps [45]. Cardiorespiratory fitness testing was conducted using a submaximal exercise test based on a modified Astrand Rhyming protocol with an adaptive ergometer to generate VO_2__peak_ values. This fitness test previously demonstrated excellent reliability for VO_2__peak_ values at maximal steady state (ICC 0.98) [46]. 

### 2.6. Data Analysis

Data were analysed using IBM SPSS Statistics, Chicago, US (version 24). Summary statistics described participant characteristics. The unadjusted prevalence rates of adherence/non-adherence with smoking abstinence, dietary intake of 5 portions of fruit and vegetables per day and minimal physical activity participation of 30 min of moderate–vigorous activity three times per week were extrapolated from the self-reported questionnaires, reported as the count and percentage. Adherence with all three behaviours was similarly reported. Chi squared tests explored differences in prevalence rates by gender, stroke/TIA classification and stroke aetiology (TOAST). Independent t-tests examined whether there were differences in time since the stroke event between those adherent and those non-adherent with each health behaviour. To evaluate the predictive ability of multiple factors identified in the conceptual model, independently of the other factors, multiple logistic regression models for adherence versus non-adherence with each health behaviour and all three behaviours were created. Non-adherence was coded as 0, and adherence was given the value 1. Each model included seven independent variables. The explanatory power of each model was determined by the value of correct classification (the percentage of participants correctly classified as being compliant with the dependent health behaviour from the information given by the independent variables included in the model). The required 2-tailed significance level for all tests was set at 0.05. Limited responses in TOAST classification levels 4 and 5 resulted in these variables being collapsed to one variable, and subsequently removed in the regression models due to low values.

## 3. Results

Participant characteristics (*N* = 142) are summarised in Table 1. The sample included individuals with both stroke and TIA, mean age 63 years and 70% were male. Mean blood pressure values indicate good control of hypertension in the sample. BMI values classify the group as overweight. However, 35% of participants met the threshold for the category of obese. 

Adherence rates with healthy lifestyle recommendations for secondary prevention of smoking abstinence, MVPA participation, high fruit and vegetable consumption, and adherence with all three are detailed in Table 2. Higher proportions of participants with TIA were noted to meet the physical activity recommendations. No differences were identified in time since stroke between those who were adherent and those who were not with the health behaviours tested. 

Logistic regression models addressing predictor variables for adherence with stroke secondary prevention health behaviours are summarised in Table 3. Each model containing all seven predictor variables could distinguish between those adherent to the health behaviour/s and those who were not, with the exception of the model where fruit and vegetable consumption was the dependent variable. With non-smoking status as the dependent variable, the full model explained between 19% (Cox and Snell R square) and 32% (Nagelkerke R squared) of the variance in adherence with smoking abstinence, correctly classifying 88% of cases. Two independent variables (psychological distress and age) made a statistically significant contribution to the model. 

With adherence to fruit and vegetable consumption recommendations, the full model explained between 11% and 14% of the variance in adherence with the dietary requirements correctly distinguishing 66% of cases. Only gender made an independent and unique contribution to the model.

The model exploring adherence with MVPA explained between 17% and 24% of the variance in MVPA participation, correctly classifying 74% of cases. Only cardiovascular fitness (VO_2__peak_) made a unique and significant contribution to the model. 

Finally, the model examining determinants of combined adherence with the three health behaviours explained between 16% and 26% of the variance in the dependent variable, correctly classifying 83% of cases. Only VO_2__peak_ values made a unique and significant contribution to this model. 

## 4. Discussion

This study described the prevalence of adherence with health behaviours for risk reduction after stroke and examined a range of predictor variables, based on an a priori conceptual model. Despite the study participants’ raised risk for future stroke events [3,4], the overall proportion adherent with lifestyle-related recommendations addressing smoking, MVPA participation and fruit and vegetable consumption was found to be suboptimal. While the health behaviours reported and the proportion of study participants categorised as overweight and obese by BMI category in this study were equivalent to the available population normative data for a similar age range [47], this study suggests that little behaviour change occurred following stroke and presents clear opportunities to enhance secondary prevention after stroke. The levels of adherence with individual behaviours of smoking abstinence, fruit and vegetable consumption and MVPA observed in this study are in line with those reported in other studies after stroke [26,30,32,33,48]. Where the regression models employed in the current study identified predictors of one or more lifestyle-related, modifiable risk factor for recurrent events, the authors discuss these in detail below and further present a proposed decision tree for interventions to guide clinicians based on the findings of this study (Figure 2).

The prevalence rate for adherence with three common health behaviour recommendations observed in this study was strikingly low at 18%. Combined health behaviours are recognised to strongly influence successful aging [49]. In individuals after stroke, higher combined health behaviours are also shown to decrease cardiovascular and all-cause mortality, with the cumulative effect observed highlighting a dose dependent response to combining health behaviours after stroke [50]. Therefore, the low combined adherence rate identified in this study presents a clear target for enhanced stroke secondary prevention. Five-year rates of stroke recurrence fell from 18% to 12% between 1999 and 2005 but further reductions in subsequent years were not apparent as compliance with physiological targets and secondary prevention medication prescription became better optimised [51]. Yet, many more unmanaged modifiable risk factors occur in people with stroke [52] and this study offers new insights into poor adherence in combined health behaviours that warrant consideration in targeted interventions to further enhance stroke secondary prevention efforts. 

Insufficient levels of moderate–vigorous physical activity (typically < 10 MET hours/week) have been identified as predictive of recurrent ischaemic stroke [53]. From the range of predictor variables tested in this study, cardiovascular fitness presents the most utility as a determinant, not just for physical activity participation but for combined health behaviours after stroke, presenting a legitimate target in stroke secondary prevention. Cardiorespiratory fitness was previously identified as having a high association with post stroke physical activity levels in a systematic review [54]. This study now identifies that for each VO_2__peak_ incremental increase in L/min, the person with stroke is 7.5-fold more likely to adhere to the MVPA recommendations. The need for aerobic training interventions after stroke is clear as cardiorespiratory fitness levels are roughly half that of age-matched sedentary counterparts and are often insufficient to meet the threshold level required for basic activities of daily living [55,56,57,58]. Furthermore, after stroke, individuals make cardiorespiratory adaptations to aerobic exercise comparable to those of healthy, sedentary adults and to cardiac participants in exercise-based cardiac rehabilitation programmes, where improvements in the range of 2–3.6 METs are reported [59,60]. Cardiovascular fitness as a vehicle for health behaviour adoption across multiple behaviours also warrants careful consideration in stroke secondary prevention based on the findings in this study which indicate that people are 12-fold more likely to be adherent to recommendations across three health behaviours for each incremental improvement in VO_2__peak_. A key property of health risk behaviours is that they co-occur as clusters or bundles [61]. In the behavioural sciences, symmetrical changes resulting from health promotion interventions are not assumed although it is acknowledged that many lifestyle behaviours are served by shared, self-regulatory resources implying that when intervening upon one health behaviour, corollary changes in other health behaviours may occur [62]. Evidence supporting the role of an aerobic-based exercise intervention to affect change in other health behaviours is not clear and conflicting results related to its protective effect against smoking as well as a supportive effect on smoking cessation treatments exist [61,62,63,64,65]. 

In exploring determinants of healthy eating after stroke, the conceptual model proposed did not adequately explain adherence with daily fruit and vegetable consumption recommendations but did note males were 0.41-fold less likely to meet this recommendation. The proportion (54%) of individuals after stroke meeting this recommendation is lower than that reported in a study based in China at 60% [66], suggesting that culture and ethnicity may play a role in this health behaviour. The influence of environment, social norms and family members in health behaviours including fruit and vegetable intake is well established [67,68]. Qualitative synthesis of studies addressing lifestyle after stroke previously identified family and carers’ ability to exert both positive and negative influences on behavioural patterns [69] and a focus group study examining barriers to healthy lifestyle after stroke identified that, particularly in men, the person with stroke often did not buy or prepare their own food/meals [35]. Inclusion of family members/carers in dietary change after stroke may be important in achieving sustainable, behaviour change and future models of care should explore this aspect in more depth. 

This study identified that for each point reduction in the combined HADS scale for psychological distress, a person was 0.90-fold less likely to smoke after stroke. This study did not report the smoking cessation rates immediately after stroke where registry data identify that most change in smoking status occurs within three months after stroke [26,31]. Given the time elapsed from stroke or TIA for participants in the current study, the determinants identified most likely represent those relevant to persistent smokers in the chronic phase after stroke and do not reflect all smokers immediately after a stroke event. This distinction may in part explain sex differences identified in persistent smoking after stroke identified in comparison to the literature [26]. For persistent smokers following stroke, screening for and addressing the management of psychological distress levels, where present, may prove an important component of future successful smoking cessation interventions. The effects of stroke (including depression) have previously been identified in qualitative synthesis to affect agency over lifestyle issues after stroke [69] and people with a lifetime history of depression are twice as likely to be smokers and require additional support and longer courses of treatment to achieve smoking cessation [70]. Further research is required in this area. 

Study limitations apply that include the relatively small number of datasets employed in the analysis, the higher proportion of males and that participants were individuals who self-selected to an exercise and lifestyle counselling based RCT [36]. This may represent a sub-population of stroke that are more receptive to lifestyle change and exercise participation. However, similar findings with prevalence rates of health behaviours after stroke reported in other studies/registry-based data suggest that they are broadly representative of stroke survivors in their health behaviours [26,30,32,33]. Similarly, the significance of the different factors considered in the current study could vary in different countries (e.g., low and low-middle income countries) and thus results may not extrapolate to all post-stroke populations. A number of other contextual factors should be considered in the interpretation of data presented. Cross-sectional data were used to determine the odds of adherence with risk-reducing behaviours. However, future prospective studies after stroke are required for confirmation of our findings. Most recurrences in stroke remain largely unexplained by conventional risk factors [71] and while the conceptual model adopted used previously identified person and stroke-related factors shown to be predictive of physiological outcomes and stroke recurrence, not all factors that could influence the adherence to healthy lifestyle post-stroke were considered. The model did not, as identified, consider environmental and social factors (norms and control) in identifying predictors of adherence with stroke secondary prevention lifestyle guidelines. Similarly, safe alcohol consumption, stress management, other health behaviours and BMI, all important for risk reduction after stroke, were not included in this study due to limited data available. These may be important risk factors to address in future modelling studies. The authors acknowledge the decision tree provided is not inclusive of all health behaviour risk factors and only represents those presented in the current study.

## 5. Conclusions

This cross-sectional study of individuals after ischaemic stroke or TIA identifies low levels of adherence with combined health behaviours. Opportunities for enhanced secondary prevention through targeted interventions addressing multiple risk-reducing behaviours exist. Aerobic fitness levels were the strongest predictor of both physical activity participation and combined health behaviours (addressing diet, smoking and physical activity). Fitness training interventions therefore comprise an important component of the stroke secondary prevention toolkit. Additional environmental and social factors may need to be considered in promotion of fruit and vegetable consumption, alongside the gender aspect identified. In younger individuals after stroke and/or where higher levels of psychological distress exist, additional strategies are warranted for successful smoking cessation. 

## Figures and Tables

**Figure 1 ijerph-18-04666-f001:**
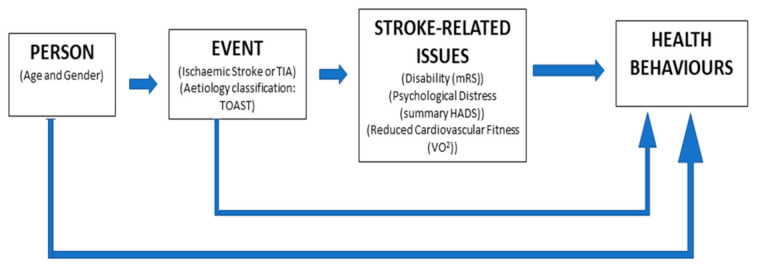
Conceptual model of pathways between determinants of health behaviours after stroke.

**Figure 2 ijerph-18-04666-f002:**
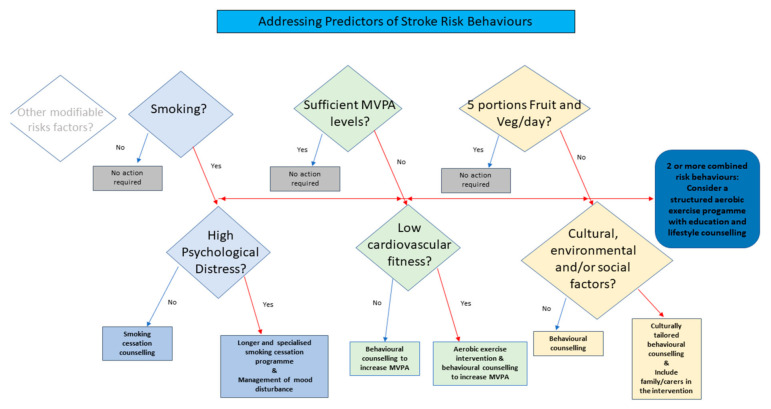
Decision tree addressing predictors of adherence to lifestyle recommendations in stroke risk management.

**Table 1 ijerph-18-04666-t001:** Participant characteristics.

		*N* = 142
		Mean (SD)
Age		63.1 (12.7)
Time since stroke/TIA (months)		37.3 (32)
BMI		28.3 (5.4)
Blood pressure (SBP/DBP mmHg)		136 (19)/82 (12)
Initial VO_2_ peak (LO_2_/kg/min)		1.3 (0.6)
		Median (range)
Modified Rankin Scale		1 (0–4)
Functional ambulatory category		5 (0–5)
		*N* (%)
Male		100 (70.5)
Female		42 (29.5)
Left CVA		54 (38)
Right CVA		42 (29.6)
TIA		46 (32.4)
TOAST classification	1	94 (66)
	2	23 (17)
	3	18 (13)
	4	01 (0.5)
	5	06 (5)

BMI: body mass index; CVA: cerebrovascular accident; TIA: transient ischaemic attack; TOAST: the Trial of Org 10172 in Acute Stroke Treatment aetiology classification system, where 1: large-artery atherosclerosis; 2: cardioembolism; 3: small-vessel occlusion; 4: stroke of other determined aetiology; 5: stroke of undetermined aetiology.

**Table 2 ijerph-18-04666-t002:** Unadjusted prevalence of compliance with healthy lifestyle guidelines for stroke secondary prevention.

	Smoking Cessation	≥5 Portions Daily Fruit and Vegetables/Week	Minimum 30 min MVPA 3/Week	Compliance with All Three Health Behaviours
	*n* (%)	*n* (%)	*n* (%)	*n* (%)
Total compliance	119 (84)	76 (54)	44 (31)	26 (18)
Male (*n* = 100)	87 (87)	49 (49)	30 (30)	17 (17)
Female (*n* = 42)	32 (76)	27 (64)	14 (33)	9 (21)
	*p* = 0.11	*p* = 0.096	*p* = 0.69	*p* = 0.53
Stroke (*n* = 95)	81 (85)	51 (54)	24 (25)	14 (15)
TIA (*n* = 47)	38 (81)	25 (53)	20 (43)	12 (26)
	*p* = 0.50	*p* = 0.96	*p* = 0.04 *	*p* = 0.12
TOAST 1 (*n* = 94)	79 (84)	49 (52)	27 (29)	17 (18)
TOAST 2 (*n* = 23)	21 (91)	15 (65)	6 (20)	4 (17)
TOAST 3 (*n* = 18)	13 (72)	9 (50)	8 (44)	2 (11)
TOAST 4 and 5 (*n* = 7)	6 (86)	3 (43)	3 (43)	3 (43)
	*p* = 0.57	*p* = 0.63	*p* = 0.48	*p* = 0.30

* Significant difference by stroke/TIA classification in proportion meeting physical activity guidelines (Chi^2^4.40).

**Table 3 ijerph-18-04666-t003:** Logistic regression models of adherence with health behaviours for stroke secondary prevention.

Health Behaviour	Non-Smoking ^1^			5 Fruit and Vegetables/Day ^2^			Minimum MVPA/Week ^3^			Meeting 3 Health Behaviours ^4^		
Independent Variables	aOR	95% CI	*p*	aOR	95% CI	*p*	aOR	95% CI	*p*	aOR	95% CI	*p*
Gender (Male)	1.06	0.26; 4.24	0.94	0.36	0.14; 0.96	0.04	0.44	0.15; 1.26	0.13	0.45	0.14; 1.53	0.2
Age	1.08	1.01; 1.16	0.03	1.01	0.97; 1.06	0.61	0.99	0.94; 1.04	0.74	1.02	0.96; 1.08	0.57
Stroke or Tia (Stroke)	1.85	0.29; 11.61	0.51	0.92	0.26; 3.23	0.90	2.15	0.53; 8.90	0.29	1.30	0.28; 5.93	0.74
TOAST			0.32			0.52			0.54			0.74
TOAST 1	0.12	0.01; 2.92	0.19	2.13	0.28; 16.31	0.47	0.96	0.11; 8.43	0.97	0.51	0.05; 4.92	0.56
TOAST 2	0.14	0.00; 5.47	0.29	4.93	0.46; 53.28	0.20	1.35	0.11; 17.12	0.82	1.01	0.07; 15.22	0.99
TOAST 3	0.05	0.00; 1.41	0.08	2.24	0.24; 21.03	0.48	2.77	0.26; 29.06	0.49	0.40	0.03; 5.24	0.49
Disability (Mrs)	1.09	0.59; 2.03	0.79	1.12	0.73; 1.72	0.62	1.20	0.70; 2.06	0.51	1.44	0.76; 2.71	0.26
Psychological Distress (Hadstotal)	0.9	0.82; 0.98	0.02	0.95	0.89; 1.01	0.10	0.94	0.87; 1.02	0.14	0.92	0.83; 1.01	0.09
Cardiovascular Fitness (VO_2peak_)	1.51	0.23; 9.70	0.67	2.76	0.74; 10.27	0.13	7.55	1.65; 34.64	0.01	12.08	2.08; 70.05	0.01

^1^ Model: Χ2 (Df9, *n* = 142) = 21.7, *p* = 0.01; ^2^ Model: Χ2 (Df9, *n* = 142) = 11.9, *p* = 0.22; ^3^ Model: Χ2 (Df9, *n* = 142) = 11.9, *p* = 0.22; ^4^ Model: Χ2 (Df9, *n* = 142) = 18.6 (Df9, *n* = 142), *p* = 0.03; TIA: transient ischaemic attack; mRS: modified Rankin Scale; HADS: Hospital Anxiety and Depression Scale; MVPA: moderate–vigorous physical activity; aOR: adjusted odds ratio.

## Data Availability

Data available on request due to restrictions e.g., privacy or ethical. The data presented in this study are available on request from the corresponding author. The data are not publicly available as specific consent was not sought from participants for this purpose.

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
