# Peer review of "Predictors of Adherence to Lifestyle Recommendations in Stroke Secondary Prevention"

_ijerph, 2021, doi:10.3390/ijerph18094666_

Round 1
Reviewer 1 Report
Dear Authors,
I would first congratulate you for your work.
I have the following remarks/recommendations that to my opinion will improve the quality of your work:
Line 2, the title: I would change words’ place like this “…secondary stroke prevention”. It sounds better.
Line 30: "……a lifestyle related cardiovascular condition…” must be omitted because it is not quite true if we mean all possible etiologies of ischemic stroke
Please, make the introduction more succinct. It has to be cut at least by half. The appropriate place of most of the text is in “Discussion”
Please, add some “Study limitations”, such as the small number of patients and the fact you have analyzed some, but not all factors that could influence the adherence to lifestyle in post-stroke patients. In addition, the significance of the different factors could be varying a lot in the different countries, for which the results cannot be extrapolated to all post-stroke patients.
Please, start "Discussion" with your own results, comparing them to the results of other authors. Shorten the discussion, make it more succinct, leaving just the “corner stone” information about this topic. Emphasize on discussing your own results.
After all, what is the practical issue of this manuscript? Please, propose (elaborate) a “strategy” (practical approach, algorithm), based on your results how to improve patient’s adherence to lifestyle. It would be the real value of this work. You can test further how this strategy works. It is known that the appropriate primary and secondary CV prevention could cut the cardiovascular death rates by 50%, while the best to this moment pharmacological and interventional treatment strategies could reduce the CV mortality by no more than 40%.
Author Response
I would first congratulate you for your work. I have the following remarks/recommendations that to my opinion will improve the quality of your work:
Response: Thank you for taking the time to review this manuscript and your well considered feedback. The manuscript is significantly improved in its overall readability and in delivering its key messages as a result of these recommendations.
Line 2, the title: I would change words’ place like this “…secondary stroke prevention”. It sounds better:
Response. I chose this order of wording as it is the preferred wording of INSsPiRe-the international network of stroke secondary prevention researchers. They consider stroke secondary prevention only covers recurrent stroke events and not other cardiovascular events. Stroke secondary prevention covers allows all cardiovascular events to be considered in secondary prevention following stroke. I would prefer to keep fidelity with their preferred wording as in their consensus statement: Lawrence M, Asaba E, Duncan E, Elf M, Eriksson G, Faulkner J, Guidetti S, Johansson B, Kruuse C, Lambrick D, Longman C. Stroke secondary prevention, a non-surgical and non-pharmacological consensus definition: results of a Delphi study. BMC research notes. 2019 Dec;12(1):1-6.
Line 30: "……a lifestyle related cardiovascular condition…” must be omitted because it is not quite true if we mean all possible etiologies of ischemic stroke.
Response: Thank you for highlighting that not all ischaemic stroke aetiologies may be attributable to lifestyle factors. The phrase “a lifestyle related cardiovascular condition” has been removed completely,
Please, make the introduction more succinct. It has to be cut at least by half. The appropriate place of most of the text is in “Discussion”
Response: Thank you for highlighting this issue with the introduction. It has been significantly revised down. The word count of the original introduction was 852; the word count now is 507. In response to reviewer 2 we were asked to include a hypothesis and aims and were unable to reduce the count further while satisfying additional requests by other reviews. We hope you find it significantly improved and an suitably adequate introduction to the topic.
Please, add some “Study limitations”, such as the small number of patients and the fact you have analyzed some, but not all factors that could influence the adherence to lifestyle in post-stroke patients. In addition, the significance of the different factors could be varying a lot in the different countries, for which the results cannot be extrapolated to all post-stroke patients.
Response: Thank you for highlighting these additional limitations to be acknowledged in the paper. I have subsumed these comments into the section addressing contextual factors for interpretation. This now reads as:
Study limitations apply that include the relatively small number of datasets employed in analysis and that they were drawn from participants by self-selection to an exercise and lifestyle counselling based RCT [40]. This may represent a sub-population of stroke that are more receptive to lifestyle change and exercise participation, however similar findings between this study and the prevalence rates of health behaviours after stroke reported in other studies and registry-based studies suggests that they are broadly representative of stroke survivors in their health behaviours [19,34,31,55, 56]. Similarly, the significance of the different factors considered in the current study could vary in different countries (e.g.low and low-middle income countries) and results may not extrapolate to all post-stroke populations. A number of other contextual factors should be considered in the interpretation of data presented in the current study. Cross sectional data were used to determine the odds of adherence with risk reducing behaviours, however future prospective studies after stroke are required for confirmation of our findings. Most recurrences in stroke remain largely unexplained by conventional risk factors [80] and while the conceptual model adopted used previously identified person and stroke related factors shown to be predictive in studies addressing physiological outcomes and stroke recurrence, not all factors that could influence the adherence to lifestyle in post-stroke patients were considered. The model did not, as identified, consider environmental and social factors (norms and control) in identifying predictors of adherence with stroke secondary prevention lifestyle guidelines, nor did it include alcohol consumption and BMI and these may be important factors to address in future modelling studies.
Please, start "Discussion" with your own results, comparing them to the results of other authors. Shorten the discussion, make it more succinct, leaving just the “corner stone” information about this topic. Emphasize on discussing your own results.
Response: The discussion section has been completely rewritten (highlighted in red in the revised manuscript). It is reduced in length by about 24% and we have pivoted discussion points to focus more directly on the results presented in this manuscript. Thank you for this guidance.
After all, what is the practical issue of this manuscript? Please, propose (elaborate) a “strategy” (practical approach, algorithm), based on your results how to improve patient’s adherence to lifestyle. It would be the real value of this work. You can test further how this strategy works. It is known that the appropriate primary and secondary CV prevention could cut the cardiovascular death rates by 50%, while the best to this moment pharmacological and interventional treatment strategies could reduce the CV mortality by no more than 40%.
Response: Again the authors thank you and other reviewers for this prompt to make the findings more accessible to practice. We have developed a decision tree to guide clinicians using the findings from this study (Figure 2 in the revised manuscript).
Reviewer 2 Report
The general idea of the investigation is good and the study design is also correct.
These are the 3 main problems of the investigation
1) Aims and hypothesis not defined
2) Repetition of the tables results in the text (too long text)
3) Lack of practical application of the results
Introduction
Line 10. Do not use initials without the full name (first appearance)
Line 16. Symbol not admitted by the SI (International system of units)
Line 74 and 75. The format of p values is different than in the abstract. Use one
Methods
Line 105. Indicate where Ethical approval was obtained (university, hospital....)
Line 130. Indicate brand and country of the instruments
Line 159. Change "is" for "was"
Results
Line 200. Improve format of the data between brackets
Line 237. Improve headings format to match columns
Discussion
Line 365. Include practical application of the results of the study
Conclusion
Line 389-383. Long sentence. Divide in 2 for a better understanding.
Line 397-398. Change the last sentence no to be the same as in the abstract
References
The font does not match the text format
Author Response
The general idea of the investigation is good and the study design is also correct.
These are the 3 main problems of the investigation
- Aims and hypothesis not defined
Response: We have reworded the last paragraph to address this issue. It now reads as:
We hypothesises that those with non-cardiac stroke sources, those with greater stroke related disability and greater psychological distress will have lower adherence levels. Therefore, the aims of this study are to:
- present an a priori conceptual model of potential factors influencing health behaviours after stroke or TIA (Figure 1). These include sociodemographic factors (age and gender), stroke-related factors (TOAST classification, stroke or TIA classification) and stroke sequalae (acquired disability, reduced cardiovascular fitness and psychological distress.
- Establish unadjusted prevalence rates of adherence with healthy lifestyle behaviours of absolute smoking abstinence, a minimum of 30 minutes of moderate to vigorous intensity activity (MVPA) at least 3 days per week, consumption of greater than or equal to 5 portions of fruit and vegetables per day and adherence with all three behaviours.
- Identify determinants of each healthy lifestyle behaviour and adherence with all three from those proposed in the conceptual model (Figure 1).
- Repetition of the tables results in the text (too long text)
Response: the detail has been substantially revised down in the results section as highlighted by tracked changes in the document. Much of the text relating to the models have been included in the subscript of the revised table 3.
- Lack of practical application of the results
Response: Thank you and other reviewers for this prompt to make the findings more accessible to practice. We have developed a decision tree to guide clinicians using the findings from this study (Figure 2 in the revised manuscript). We believe this makes a new and meaningful contribution to the revised manuscript.
Introduction
Line 10. Do not use initials without the full name (first appearance)
Response: this now reads as transient ischaemic attack (TIA
Line 16. Symbol not admitted by the SI (International system of units)
Response: I have reviewed a number of other publications in this journal in relation to how aerobic capacity as VO2peak is reported (e.g. 10.3390/ijerph17238732 ). I have revised the abstract to read as cardiovascular fitness (V̇O2peak) measured as peak oxygen uptake during incremental exercise (L/min). I hope this is satisfactory.
Line 74 and 75. The format of p values is different than in the abstract. Use one
Response: I have used the p from the abstract consistently throughout the document now. However these particular lines were significantly revised in shortening the introduction as advised by another reviewer and the values to which you refer have been removed in response to a request to significantly shorten the introduction.
Methods
Line 105. Indicate where Ethical approval was obtained (university, hospital....)
Response: these items are detailed in the mandatory Institutional Review Board Statement at the end of the document. I had previously removed them from this section to avoid duplication. I have reworded now to signpost readers to this section. It now reads: The study received full ethical approval as detailed in the Institutional Review Board Statement.
Line 130. Indicate brand and country of the instruments:
Response: this has been included now- Body mass and stature were measured using a calibrated weighing scale with stadiometer (SECA Medical 703, Germany)
Line 159. Change "is" for "was"
Response: this has been changed. Thank you.
Results
Line 200. Improve format of the data between brackets.
Response: this has been changed and is now formatted as χ2 (DF9, N= 142) = 21.7, p =0.01. However, again in keeping with recommendations from other reviewers to revise down the results section, this text and revised format is now moved as a subscript to table 3.
Line 237. Improve headings format to match columns
Response: This table became corrupted from the original format when it was uploaded to the journal and subsequently fitted to the page limits, apologies. This table has been significantly revised in line with this comment and reviewer 3 who recommended the table be restructured. We trust it is of a higher standard on this resubmission.
Discussion
Line 365. Include practical application of the results of the study
Response: Thank you and other reviewers for this prompt to make the findings more accessible to practice. We have developed a decision tree to guide clinicians using the findings from this study (Figure 2 in the revised manuscript). We believe this makes a new and meaningful contribution to the revised manuscript. The discussion has also been revised significantly to better present the results of this study and the implications for practice.
Conclusion
Line 389-383. Long sentence. Divide in 2 for a better understanding.
Response: This now reads as: This cross-sectional study of individuals after ischemic stroke or TIA identifies low levels of adherence with combined health behaviours. Opportunities for enhanced secondary prevention through targeted interventions addressing multiple risk-reducing behaviours exist.
Line 397-398. Change the last sentence no to be the same as in the abstract. Response: I have revised the conclusion section now which reads as:
This cross-sectional study of individuals after ischemic stroke or TIA identifies low levels of adherence with combined health behaviours. Opportunities for enhanced secondary prevention through targeted interventions addressing multiple risk behaviours exist. Aerobic fitness levels were the strongest predictor of both physical activity participation and combined health behaviours (addressing diet, smoking and physical activity). Fitness training interventions therefore, comprise an important component of the stroke secondary prevention toolkit. Additional environmental and social factors may need to be considered in promotion of fruit and vegetable consumption, alongside the gender aspect identified. In younger individuals after stroke and/or where higher levels of psychological distress exist, additional strategies are warranted for successful smoking cessation.
References
The font does not match the text format:
Response: I have changed the font to match the overall article. The justification on the page is determined by the template provided by the journal.
Reviewer 3 Report
Introduction
- Well described the background and study rationale.
Methods
- Provide the ethics approval number & committee that approved it.
- Instruments used and statistical analysis were well explained
- Move the placing of Table 1 to Results section
Results
- Table 2 - very busy table which could have been better presented with p values included within the table. Consider redoing this table.
- Table 3 - this is not the conventional way of presenting log reg analysis. Consider redoing this to a more common way of presentation. Current presentation is very confusing to readers.
Discussion
- No further comments
Author Response
Thank you for taking the time to review this manuscript and your considered feedback. The manuscript is significantly revised in light of your comments/recommendations and those of the other 3 reviewers. Overall we think responding to such constructive feedback has improved the readability and the delivery of the manuscript’s key messages.
Introduction
- Well described the background and study rationale. Thank you
Methods
- Provide the ethics approval number & committee that approved it. Response: these items are detailed in the mandatory Institutional Review Board Statement at the end of the document. I had previously removed them from this section to avoid duplication. I have reworded now to signpost readers to this section. It now reads: The study received full ethical approval as detailed in the Institutional Review Board Statement.
- Instruments used and statistical analysis were well explained. Thank you
- Move the placing of Table 1 to Results section Response: This has now been placed directly in the results section as advised
Results
- Table 2 - very busy table which could have been better presented with p values included within the table. Consider redoing this table. Response: This table has now been transposed and simplified. p values have been included as requested.
- Table 3 - this is not the conventional way of presenting log reg analysis. Consider redoing this to a more common way of presentation. Current presentation is very confusing to readers. Response: This has now been revised. While previously formatted as guided by Pallant; I have revised it and kept fidelity with other tables of logistical regression, previously published in this journal (e.g. Bert et al, 2020). I trust this has improved its readability.
Discussion
- No further comments
Reviewer 4 Report
This manuscript is well-written, results are clearly presented and discussed, and provide useful insights for practitioners who promote behavior change among stroke patients. I have just a few comments for consideration that occurred to me as I read the manuscript.
Authors mention in the Introduction (Lines 56-61) and Discussion(Lines 254-260) that there are five modifiable behaviors that have been associated with stroke prevention and cardiovascular mortality. It would be helpful to indicate why the three behaviors in the model were selected and why alcohol consumption and BMI were not included, perhaps in the Discussion. The sample includes a high percentage of males, who tend to drink alcohol more often, (and also tend to eat fewer fruits and vegetables than females).
Were there differences in time since stroke/TIA among adherent and non-adherent individuals? – this appears to have varied considerably among the sample. If there is any data on effect of time on adherence, it might be helpful to include this in the discussion as a factor for consideration for promoting adherence. Is it possible that people are more adherent in the time soon after a stroke?
It is not clear to me why etiology of stroke/TIA would be a factor influencing adherence to lifestyle recommendations.
Author Response
This manuscript is well-written, results are clearly presented and discussed, and provide useful insights for practitioners who promote behavior change among stroke patients. I have just a few comments for consideration that occurred to me as I read the manuscript.
Response: Thank you for taking the time to review this manuscript and your considered feedback. The manuscript is significantly revised in light of your comments/recommendations and those of the other 3 reviewers. Overall we think responding to such constructive feedback has improved the readability and the delivery of the manuscript’s key messages.
Authors mention in the Introduction (Lines 56-61) and Discussion(Lines 254-260) that there are five modifiable behaviors that have been associated with stroke prevention and cardiovascular mortality. It would be helpful to indicate why the three behaviors in the model were selected and why alcohol consumption and BMI were not included, perhaps in the Discussion. The sample includes a high percentage of males, who tend to drink alcohol more often, (and also tend to eat fewer fruits and vegetables than females).
Response: Thank you for pointing out these issues. We have addressed them in the manuscript now. The proportion of males is now considered in the limitations section as shown below:
Study limitations apply that include the relatively small number of datasets employed in the analysis, the higher proportion of males and that participants were individuals who self-selected to an exercise and lifestyle counselling based RCT [40].
The additional modifiable behaviours that were not addressed in the study are also discussed in the limitations section as shown below:
Similarly safe alcohol consumption, stress management, other health behaviours and BMI, all important for risk reduction after stroke, were not included in this study due to limited data available. These may be important risk factors to address in future modelling studies. The authors acknowledge the decision tree provided is not inclusive of all health behaviour risk factors and only represents those presented in the current study.
Were there differences in time since stroke/TIA among adherent and non-adherent individuals? – this appears to have varied considerably among the sample. If there is any data on effect of time on adherence, it might be helpful to include this in the discussion as a factor for consideration for promoting adherence. Is it possible that people are more adherent in the time soon after a stroke?
Response: We conducted additional analysis to address this question. No statistically significant differences in time since stroke event were identified for this item. This is now addressed in the analysis and results sections. I further address in the discussion section that most smoking cessation takes place early after stroke and I address the limitations of our study findings in this regard.
This says that after the first year very few people stop taking medication; Hamann GF, Weimar C, Glahn J, Busse O, Diener HC. Adherence to secondary stroke prevention strategies–results from the German Stroke Data Bank. Cerebrovascular diseases. 2003;15(4):282-8.
It is not clear to me why etiology of stroke/TIA would be a factor influencing adherence to lifestyle recommendations.
Response: Cardioembolic sources of stroke in particular may not be as associated with the more traditional lifestyle related factors and these individuals may have healthier lifestyles than those who have had an ischaemic stroke related to cardiovascular disease. For example the source of stroke may may be associated with valvular disease. The aetiology of stroke is thus included as it is identified in the literature, as addressed in the introduction, as a predictor of stroke recurrenence and for these reasons we included it in the analysis. We acknowledge that this may not have been clearly stated and have addressed this in more detail as follows:
Pathophysiological factors, stroke subtype (aetiology) and lifestyle factors have been used to identify people at risk for recurrent events [25-27]. However, longitudinal studies addressing recurrence lack specific focus on lifestyle factors [28-30] and some stroke types e.g. cardioembolic stroke may not be as strongly linked to established behavioural risk factors.
We also now specifically address this in the hypotheses we were asked to include:
We hypothesised that those with non-cardiac stroke sources, those with greater stroke related disability and greater psychological distress will have lower adherence levels.
Round 2
Reviewer 1 Report
The current mansucript presents the results of a cross-sectional study with 142 post-stroke patients, exploring factors that could exert influence on patient's adherence to healthy lifestyle measures/behaviour. Based on their results the authors have tried to elaborate a decision-making algorithm, improving the secondary stroke prevention.
After author's revision the mansucript has been significantly improved compared to the original version. The authors have observed most of the reviewers recommendation.
To my oponion, the manuscript can now be considered for publication in its current form.